# Flexoelectric domain walls enable charge separation and transport in cubic perovskites

Dmytro Rak [1,2], Dusan Lorenc[1,5], Daniel M. Balazs [1], Ayan A. Zhumekenov [3,6], Osman M. Bakr [4] & Zhanybek Alpichshev [1] ✉

The exceptional energy-harvesting efficiency of lead-halide perovskites arises from unusually long photocarrier diffusion lengths and recombination lifetimes that persist even in defect-rich, solution-grown samples. Paradoxically, perovskites are also known for having very short exciton decay times. Here, we resolve this apparent contradiction by showing that key optoelectronic properties of perovskites can be explained by localized flexoelectric polarization confined to interfaces between domains of spontaneous strain. Using birefringence imaging, electrochemical staining, and zero-bias photocurrent measurements, we visualize the domain structure and directly probe the associated internal fields in nominally cubic single crystals of methylammonium lead bromide. We demonstrate that localized flexoelectric fields spatially separate electrons and holes to opposite sides of domain walls, exponentially suppressing recombination. Domain walls thus act as efficient mesoscopic transport channels for long-lived photocarriers, microscopically linking structural heterogeneity to charge transport and offering mechanistically informed design principles for perovskite solar-energy technologies.

Halide perovskites have come forward to the limelight as prospective future-generation photovoltaic materials, offering significant advantages over conventional technologies, such as low manufacturing costs and solution processability. The efficiency of the perovskite solar cells has improved tremendously over the last decade and has now reached 27%[1,2] for single-junction cells. Lead-halide perovskites (LHPs) are particularly interesting for solar energy harvesting applications. The remarkable photovoltaic performance of these materials is a consequence of a combination of factors, such as high optical absorption coefficients[3,4], high defect tolerance[5–7], and long charge carrier lifetimes[6,8] and diffusion lengths[6,9]. However, despite intensive research, the nature of the optoelectronic properties underlying such astounding performance is unclear.

LHPs, such as $MAPbI_3$ and $MAPbBr_3$ ($MA = CH_3NH_3^+$), have been thoroughly studied to understand the nature of their superior optoelectronic properties. It was realized early on that the diffusion coefficients in LHPs are not remarkable at all, and that the exceptionally long diffusion lengths critical for photovoltaic applications are mainly due to unusually long photocarrier recombination times (e.g., ref. [6]). Several explanations have been proposed, including strong spin-orbit coupling leading to Rashba splitting[10–12], a bulk photovoltaic effect resulting from ferroelectric ordering[13–16], dynamical symmetry lowering through fluctuations[17], or reduced scattering of charge carriers protected by the formation of large polarons[18,19]. Particular attention was paid to the possibility of ferroelectric ordering, which would explain the long lifetimes and diffusion lengths of the charge carriers.

[1]Institute of Science and Technology Austria, Klosterneuburg, Austria. [2]Institute of Experimental Physics, Slovak Academy of Sciences, Košice, Slovakia. [3]King Abdullah University of Science and Technology, Thuwal, Kingdom of Saudi Arabia. [4]Center for Renewable Energy and Storage Technologies (CREST), Division of Physical Science and Engineering (PSE), King Abdullah University of Science and Technology, Thuwal, Kingdom of Saudi Arabia. [5]Present address: International Laser Centre, Ilkovicova 3, Bratislava, Slovakia. [6]Present address: School of Materials Science and Engineering (MSE), Nanyang Technological University, Singapore, Singapore. ✉e-mail: alpishev@ist.ac.at

The idea of ferroelectric ordering is generally tempting, considering the oxide perovskites are well-known ferroelectrics. However, most structural studies of MAPbI$_3$ and MAPbBr$_3$ do not support this hypothesis. At room temperature, their space groups were identified[20–22] as non-polar tetragonal $I4/mcm$ and cubic $Pm\bar{3}m$, respectively, with ferroelectricity forbidden by inversion symmetry. At the same time, some studies identified the structure of tetragonal MAPbI$_3$ as the polar group $I4cm$[23,24]. Recent studies have presented compelling evidence of ferroelectric ordering in the tetragonal phases of MAPbI$_3$[14,15] and MAPbBr$_3$[16], making the puzzling nature of this phase a topic of extensive debate. Meanwhile, the cubic phase has received relatively less attention, despite the tetragonal-to-cubic phase transition having little effect on the performance of MAPbI$_3$ in solar applications[25], indicating that the mechanism underlying its unique properties remains unaffected.

In this paper, we investigate the optoelectronic properties of the room-temperature polymorph of MAPbBr$_3$, previously identified as cubic. Our key finding is the discovery of local flexoelectric polarization confined to the boundaries between microscopic domains of unequal strain, resulting in local electric fields that generate long-lived photocurrents under zero bias. The spatial separation of photocarriers at the flexoelectrically-polarized domain walls and the consequent exponential suppression of their recombination naturally explain many previously reported anomalies in the charge dynamics of LHPs. We confirm the complex domain structure of the nominally cubic phase by visualizing the domain walls using an original electrochemical staining technique. Our results provide mechanistic insight into the structural complexity of lead-halide perovskites and establish a framework for designing and optimizing their application in photovoltaic technology.

## Results

### Birefringence in cubic MAPbBr3

Our investigation begins with the observation of an unexpected optical anisotropy in the room-temperature phase of MAPbBr$_3$. Similar to other LHPs, the basic structure of MAPbBr$_3$ is composed of corner-sharing BX$_6$ (Fig. 1a) octahedra, forming transparent rectangular-shaped crystals with a nominally cubic symmetry above $T \approx 230$ K[26]. However, a careful study of generic crystals demonstrates this is not the case in general. Fig. 1b shows a typical pristine MAPbBr$_3$ crystal grown from solution by inverse temperature crystallization[7]. To examine its optical properties, the crystal was placed in a rotating crossed-polarizer setup (Fig. 1c) and illuminated by a 632.8 nm He-Ne laser beam normal to the (001) plane, while varying the angle of polarization relative to the [100] crystal axis (Methods). The rotation of the crossed polarizer was captured on video (Supplementary Movie 1). Fig. 1d shows transmitted light images of the MAPbBr$_3$ crystal, illuminated with light polarized at 0° and 45°, respectively, revealing its complex birefringent structure. This is an unexpected observation, as MAPbBr$_3$ is expected to be in a cubic phase at room temperature[26], which is incompatible with the observed birefringence. This definitively indicates that the crystal symmetry of the MAPbBr$_3$ sample is lower than the nominal $Pm\bar{3}m$ space group. While the birefringence of nominally cubic MAPbBr$_3$ is commonly observed, this incongruity is not systematically addressed in the literature. It is typically attributed to the inferior quality of the solution-processed crystals[27], but the exact nature of this symmetry lowering remains to be understood.

The cubic-to-tetragonal transition in MAPbBr$_3$ is identified as a close-to-second-order first-order displacement-type phase transition driven by the rotation of BX$_6$ octahedra[21,28] (see inset to Fig. 1e). Reflecting the structural change, the electric susceptibility tensor is also modified at the phase transition, which is manifested in the corresponding change in the optical refractive index ellipsoid. Birefringence, therefore, provides information on the onset of phase transition and can be used as an indicator of the order parameter of the phase transition. X-ray diffraction studies revealed that the order parameter of the cubic-to-tetragonal phase transition in MAPbBr$_3$ is the rotation angle of the PbBr$_6$ octahedra[21]. The idea that the same order parameter drives the formation of the distorted cubic phase at room temperature seemed plausible, which is why we proceeded with measuring the temperature dependence of birefringence across the phase transition. The critical behavior was evaluated by measuring the temperature dependence of the Stokes parameters of a polarized near-infrared beam passing through the MAPbBr$_3$ crystal placed inside the cryostat (Methods, Supplementary Fig. 1, Supplementary Note 1), and fitting the cumulative phase retardation acquired by the beam to a power law $\Delta\Theta \propto |T - T_c|^{\alpha}$ (Fig. 1e). As can be seen in the figure, the temperature dependence of birefringence retains the sharp anomaly at $T_c$, with high-temperature birefringence providing a simple offset to the critical birefringence associated with the phase transition. This indicates that the symmetry lowering of MAPbBr$_3$ at room temperature has a nature distinct from the low-temperature rotation of PbBr$_6$ octahedra in the tetragonal phase.

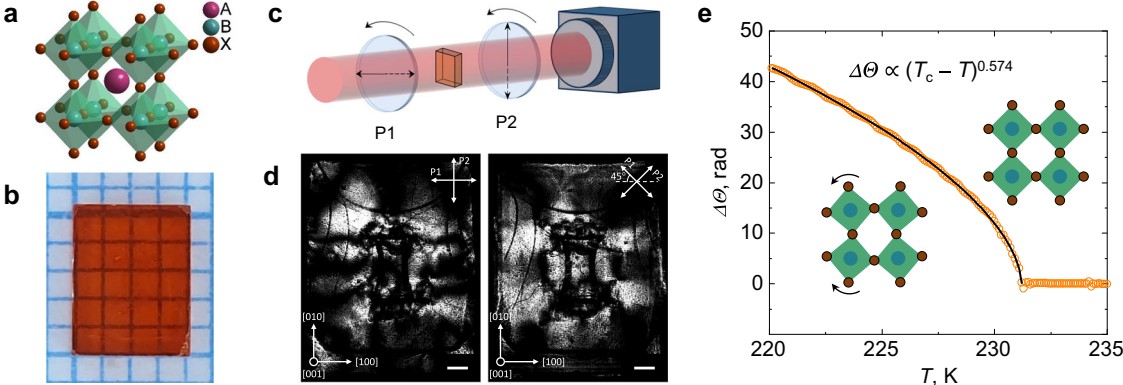

**Fig. 1 | Birefringence in the nominally cubic high-temperature phase of MAPbBr$_3$. a** Crystal structure of cubic perovskite with the general chemical formula ABX$_3$. **b** A typical MAPbBr$_3$ crystal grown from solution using the inverse temperature crystallization technique. **c** Schematic of the crossed-polarizer setup used to visualize the birefringence in the high-temperature phase of MAPbBr$_3$. **d** Polarized light images of the MAPbBr$_3$ sample shown in Fig. 1b placed between crossed polarizers oriented at 0° and 45° relative to the [100] crystal axis. The scale bars are 500 μm. **e** Cumulative phase retardation $\Delta\Theta$ as a function of temperature across the cubic-to-tetragonal phase transition in MAPbBr$_3$ single crystal. The solid black line represents the best fit to the data above the nominal critical temperature $T_c = 231.2$ °C of the form $\Delta\Theta \propto |T_c - T|^{\alpha}$. The goodness-of-fit is $R^2 > 0.999$. Inset images show the tilting pattern of PbBr$_6$ octahedra associated with the phase transition.

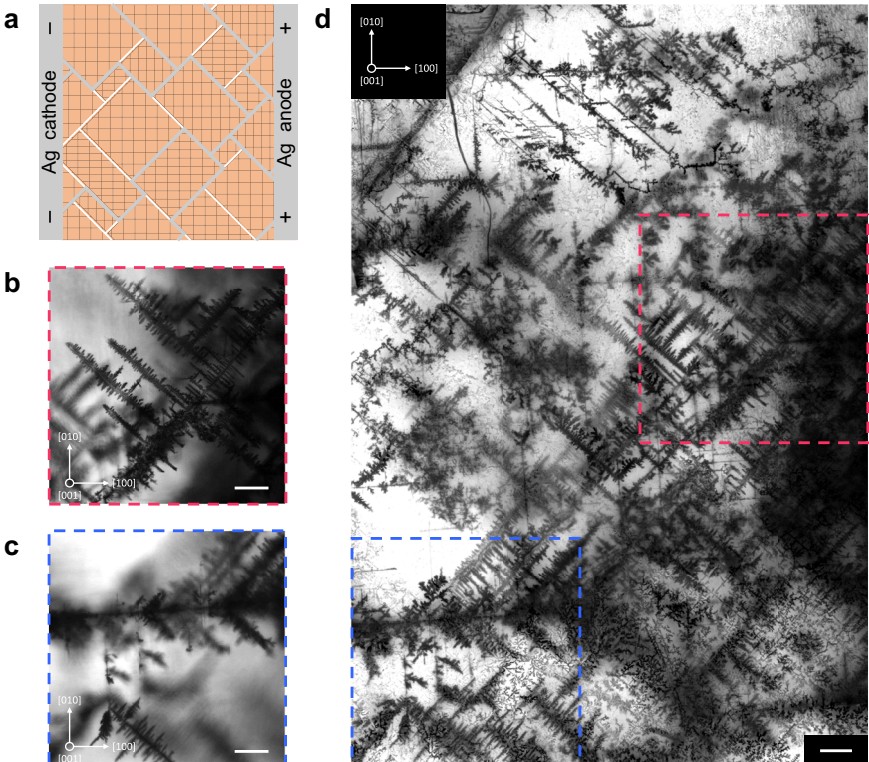

**Fig. 2 | Electrochemical visualization of the domain structure in MAPbBr₃.**
**a** Schematic of the experiment. The silver anode attached to the side of the sample serves as a source of silver ions. When an electric field is applied, the ions are electrophoretically injected into the crystal. The domain structure is revealed through the electrochemical reduction of silver ions to metallic silver (grey), making the domain walls (white) visible. The hatched areas represent domains with different orientations of strain. **b, c** Bright-field images of silver dendrites formed in MAPbBr₃ monocrystal after an electric field of 24 V mm⁻¹ was applied for 4 h. The images show single optical slices taken at 270 μm (**b**) and 365 μm (**c**) beneath the sample surface. The scale bars are 100 μm. **d** Composite image with an extended depth of field, created by focus stacking 167 bright-field images. The dashed lines mark the locations where the corresponding single optical slices displayed in (**b**) and (**c**) were taken. The scale bar is 100 μm.

## Microscopic structure of local anisotropy in MAPbBr₃

A birefringent sample can appear dark when placed between two orthogonally crossed polarizers only if the polarization axis of one of the polarizers matches the optical axis of the sample. In contrast, an optically active sample will always appear bright, while an isotropic sample will always be dark. Our optical measurements showed that no part of the sample remains permanently bright or dark for all orientations of the mutually crossed polarizers relative to the crystal axes (Supplementary Movie 1). The data indicate that the sample is birefringent everywhere, with no optical activity, and with locally defined optical axes seemingly uncorrelated with crystal directions.

Birefringence in a nominally cubic system indicates the presence of bulk strain. In a free-standing crystal, inhomogeneous strain implies the presence of defects, such as dislocations, or, in the case of spontaneous strain (ferroelasticity), domain boundaries[29]. To investigate the nature of nonuniform strain in bulk MAPbBr₃, we electrophoretically inject silver ions into the sample, which in the course of diffusion preferentially cluster near defect sites (Methods, Supplementary Note 2). Thus, when ions are ultimately reduced to metallic silver, the defects become visible under a microscope (Fig. 2a). Unlike conventional surface-probe techniques, this method provides direct access to the intrinsic structural features of the bulk material, unaffected by surface-related effects.

Fig. 2b, c show the silver-stained defect patterns in a typical solution-grown MAPbBr₃ monocrystal taken near a silver electrode attached to the (100) face of the crystal (Supplementary Fig. 2a). Dendritic structures clearly indicate domain wall patterns oriented at 45° and 90° relative to the crystal axes, suggesting the presence of 90° and 180° domain walls. Domains as small as 5×5 μm² were identified

(Supplementary Fig. 2b). Fig. 2d shows an image of the silver-stained sample generated by the focal plane merging of the bright-field Z-stack, revealing a complex domain structure of the distorted cubic phase. Supplementary Movies 2 and 3 show a complete scan along the Z-axis and the reconstructed 3D structure of silver dendrites, respectively.

To evaluate the potential effects of an applied electric field (e.g., ion migration) on the intrinsic domain structure of MAPbBr₃, we performed time-dependent electrochemical staining experiments (Supplementary Fig. 3, Supplementary Note 2). To that end, the electric field was applied in successive steps, and microscopic images of the sample were taken after each step. The results show that domain walls visualized at the onset of the staining process remain unchanged under repeated application of the electric field, confirming that staining reveals pre-existing domain wall patterns without altering them. We further observed that silver structures begin to disappear once the electric field is removed, which is why the microscopic images were acquired shortly after the staining process was completed. To investigate the dissolution process, we monitored the domain wall patterns in electrochemically treated samples over several days (Supplementary Fig. 4, Supplementary Note 2). During this period, the silver-stained domain walls became barely visible, indicating nearly complete dissolution of metallic silver.

Interestingly, MAPbBr₃ electrophoretically doped with silver also demonstrates a reversible photochromic effect (Supplementary Fig. 5, Supplementary Note 3). This phenomenon is likely analogous to that in silver halide-containing photochromic glasses, where photochemical reduction of silver ions leads to the formation of fine metallic silver particles that strongly absorb light[30].

Electrochemical staining indicates that non-cubicity in room-temperature MAPbBr$_3$ comes in the form of microscopic ferroelastic domains each hosting different strain uniform across the domain. Importantly, in this picture strain gradients are confined to structural defects−domain walls−thus alleviating the necessity of associated stress gradients. The few-micron domain size in MAPbBr$_3$ explains the illusion of smooth gradients of birefringence in the sample shown in Fig. 1d. The interpretation of the room-temperature phase in terms of ferroelasticity is reinforced by X-ray diffraction, which confirms the previously reported $Pm\bar{3}m$ space group and demonstrates the structural homogeneity of the sample (Supplementary Fig. 6, Supplementary Note 4).

## Bulk photovoltaic effect

Strain gradients break inversion symmetry and generally result in electric polarization in a phenomenon known as flexoelectricity[31]. In light of recent reports of large voltages generated in response to externally induced mechanical deformations in MAPbBr$_3$[32,33], one may wonder if spontaneous strain gradients in nominally cubic LHPs can also result in local electric polarization (Fig. 3a).

Here, we confirm the presence of local electric polarization in bulk MAPbBr$_3$ by detecting zero-bias photocurrent in as-grown single-crystal samples following localized photocarrier injection. To this end, an ultrafast sub-bandgap laser pulse is focused inside the sample, generating electron-hole pairs deep inside the bulk through two-photon absorption[34,35] (Fig. 3b–d, Supplementary Figs. 7–10, Supplementary Note 5). The photocurrent is picked up through two pairs (for vertical and horizontal currents) of non-metallic carbon leads attached to the sides of the sample by a lock-in amplifier in current detection mode (Methods). The measurements were performed on the same sample as in the birefringence measurements shown in Fig. 1d. Fig. 3e shows maps of the current frequency component corresponding to the laser pulse repetition rate ($f_{rep} = 1.5$ kHz) $I_{rep}$ obtained by scanning the beam across the sample. It is evident that the current $I_{rep}$ strongly depends on the location of carrier injection. Notably, the sign of $I_{rep}$ remains constant across large

sections of the sample, which are not symmetrical with respect to the midplane of the sample, indicating that the photocurrent is likely not due to dynamic flexoelectricity[36]. Instead, it can be understood in terms of electrostatic potential distribution $\phi(x)$ in a sample with domain-wall flexoelectricity as shown schematically in Fig. 3f. Here, the plateaus represent regions of constant strain and $\phi(x)$ within domains that both change abruptly at domain walls; the sign of photocurrent $I_{rep}$ is then determined by the average slope of electrostatic potential.

This picture offers a natural explanation to the conflicting phenomenology of the apparent ferroelectricity in LHPs: on the one hand, flexoelectric polarization at domain walls related to the difference in structure between domains naturally explains the observed pyroelectric phenomena at structural phase transitions (e.g., ref. 16), and the pinning of domains walls can account for hysteretic polarization under external electric field[14,15]. On the other, flexoelectric polarization remains confined to the domain walls, keeping inversion symmetry intact in bulk, in full agreement with optical second-harmonic generation experiments[37,38].

The zero-bias two-photon photocurrent was also observed in bulk MAPbI$_3$, upon optical excitation of a single-crystal sample (Supplementary Fig. 14, Supplementary Note 7). Although the exact nature of the inversion symmetry breaking evidenced by the observation of zero-bias photocurrent cannot be established without further investigation, there are indications that it may originate from flexoelectric domain walls. MAPbI$_3$ is tetragonal at room temperature and shows behavior sometimes interpreted as evidence of ferroelectric ordering[14,15]. However, ferroelectricity is inconsistent with second-harmonic generation studies[37,39,40], which confirm the presence of bulk inversion symmetry. Moreover, the photovoltaic efficiency of MAPbI$_3$ reportedly remains unaffected by the tetragonal-to-cubic phase transition[25]. These observations can be naturally explained by localized flexoelectricity present in both phases, suggesting that flexoelectric domain walls are not exclusive to MAPbBr$_3$ but may constitute a unifying origin of local inversion symmetry breaking in LHPs.

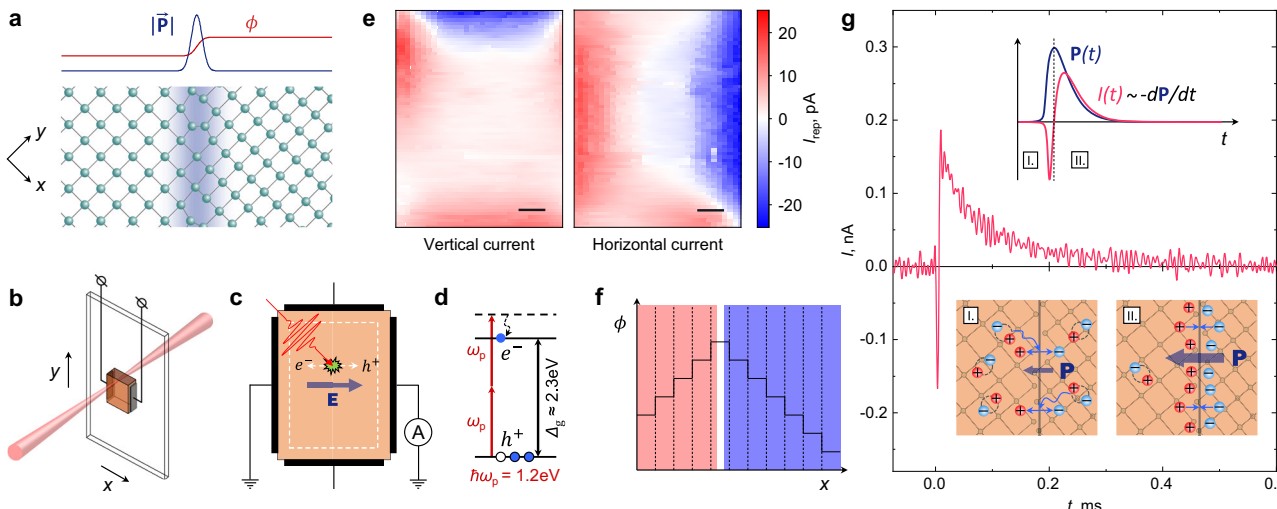

**Fig. 3 | Inversion symmetry breaking in MAPbBr$_3$ evidenced by the two-photon bulk photovoltaic effect. a** Schematic of the strain gradient confined to the domain wall and the resulting gradients of polarization **P** and potential $\phi$ induced by the flexoelectric effect. **b** Schematic of the photocurrent measurement setup. The sample can be moved relative to the beam to measure the current flowing in vertical or horizontal directions for each position of the excitation spot. **c** Schematic of the photoexcited carrier separation caused by internal electric fields. The dashed white line marks the part of the sample where the current measurements were performed. **d** Energy diagram of the two-photon absorption process. **e** Spatial distribution of the photocurrent measured in the vertical and horizontal directions. Each point represents the current measured when the sample is excited at the corresponding location. The scale bars are 500 μm. **f** Schematic of electrostatic potential distribution $\phi(x)$ in a sample with domain-wall flexoelectricity. The dashed lines represent domain walls. **g** Typical time-resolved photocurrent transient acquired in a horizontal direction. The top inset illustrates the temporal evolution of polarization **P** and current $I$ following optical excitation. The bottom inset shows schematic microscopic pictures of the charge separation and recombination phases of the photocurrent generation.

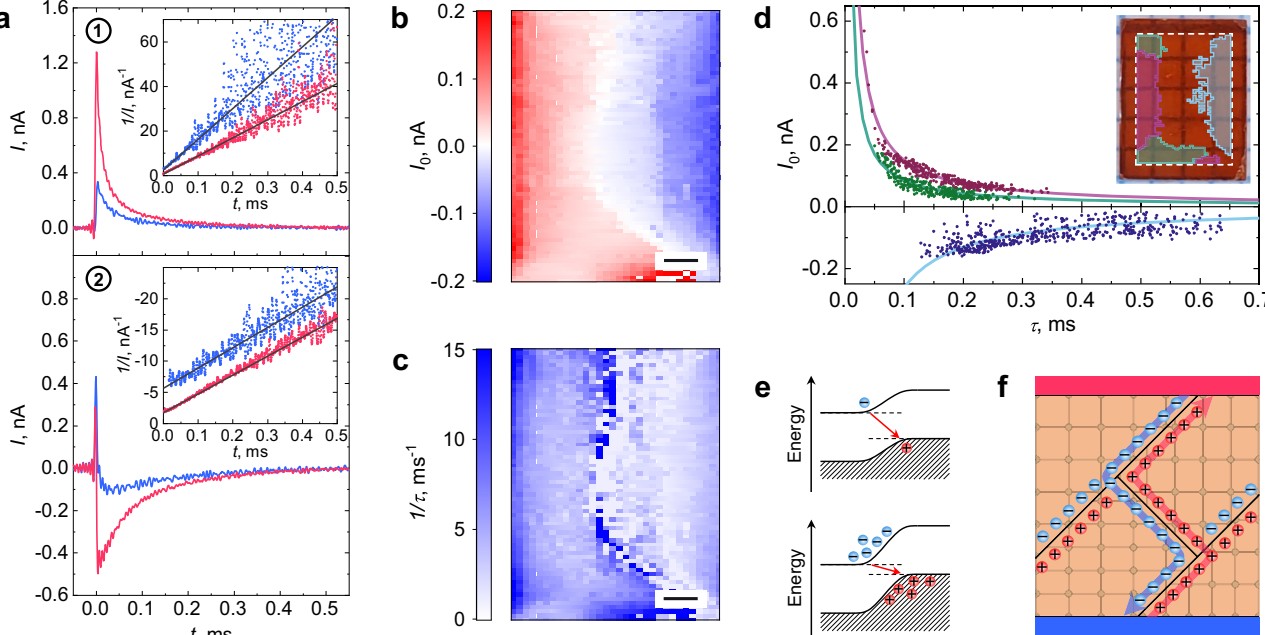

**Fig. 4 | Charge carrier dynamics in MAPbBr₃. a** Photocurrent transients measured in the horizontal direction at the respective locations indicated in (**b**). Different colors correspond to measurements performed on a sample isolated from stray light (blue) and under diffuse ambient illumination of 0.01 sun (pink), respectively. Insets show the corresponding data fitted to $1/I(t) \propto (t + const)$ (black lines). **b, c** Spatial distributions of $I_0$ and $1/\tau$, respectively, obtained from exponential fitting of current decays in regime II at respective coordinates. The scale bars are 500 μm. **d** Correlation between $I_0$ and $\tau$. Three identified data clusters are fitted with

$I_0 = C \cdot 1/\tau$, where $C$ is constant within each cluster. The inset shows a photograph of the sample, with highlighted regions corresponding to the identified clusters. The dashed white line marks the part of the sample where current measurements were taken. **e** Schematic of bandgap reduction caused by the accumulation of charge carriers at domain walls. **f** Schematic of charged domain walls acting as separate, charge-specific conductive pathways for electrons and holes in a perovskite solar cell.

## Discussion

Flexoelectric polarization at domain walls is responsible for photocurrent, but directly assigning it to simple charge diffusion along the average slopes of potential in Fig. 3f is problematic. Indeed, imagine electron-hole pairs injected, e.g., in the blue region in Fig. 3f. According to the naive picture, the holes will diffuse to the right electrode, while electrons will accumulate at the peak position of $\phi(x)$ (marked with a white stripe), which cannot be sustained indefinitely. More generally, in the absence of external bias, the current in a charge-neutral insulating sample is given by $I(t) \propto -d\mathbf{P}/dt$, where $\mathbf{P}$ is electric polarization. Therefore, if $\int I(t)dt \neq 0$ for each pulse, the total electric polarization of the sample changes monotonically throughout the course of the experiment, contradicting the observation that photocurrent in LHP samples can be generated for hours without appreciable changes in magnitude.

To interpret our observations, we first note that, based on the preceding arguments, it must be that $\int I(t)dt = 0$ or, equivalently $\mathbf{P}_{initial} = \mathbf{P}_{final}$ (see upper inset in Fig. 3g) which is in apparent contradiction with data in Fig. 3g, showing transient current response $I(t)$ reconstructed as a function of time after the laser pulse (Methods, Supplementary Fig. 11, Supplementary Note 6). We attribute this to the bandwidth limitations of the digital lock-in amplifier used for our measurements ($f \lesssim 153$ kHz), which result in the loss of high-frequency features, such as the fast spike near $t = 0$ in Fig. 3g. To confirm the displacement nature of zero-bias photocurrent, we measured the average photocurrent using high-sensitivity analog electrometer with long integration time (Supplementary Fig. 15, Supplementary Note 8). Our results show that on average there is no net charge transfer, confirming the purely displacement nature of the observed current, in full agreement with the physical picture described in the manuscript. Next, we note that although the majority of photocarriers recombine rapidly, a small fraction persists to produce the long-lived

photocurrent, whose mechanism can be understood by scrutinizing the time dependence of a typical current transient $I(t)$ as in Fig. 3g. One can identify here two distinct regimes: I) quick accumulation of polarization (a few μs) manifested as a strong and narrow initial spike in $I(t)$, on the limit of time resolution of lock-in; and II) slow relaxation of polarization (~100 μs) manifested by the slow tail in $I(t)$ opposite in sign to the initial spike, providing the main contribution to the current picked up by lock-in. Microscopically, in regime I, photocarriers diffuse to the domain boundaries where local electric fields spatially separate positive and negative charges, trapping them on the opposite sides of the domain wall; in regime II, the polarization accumulated at the domain walls is gradually relaxing via tunneling through the flexoelectric potential barrier (see bottom inset in Fig. 3g).

The tunneling-driven recombination mechanism naturally reconciles the apparent paradox between the nanosecond-scale exciton lifetimes in perovskites[41] and the millisecond-scale relaxation of photocurrent, a key factor in the photovoltaic performance of LHPs[42]. Surprisingly, in regime II, the current $I(t)$ is best described not by a simple or even bi-exponential decay (Supplementary Fig. 16, Supplementary Note 9), but rather by

$$I(t) \propto \frac{1}{t + const} \qquad (1)$$

as shown in the insets of Fig. 4a. This time dependence is further supported by an unusual correlation between the current magnitude $I_0$ (Fig. 4b) and its decay rate $1/\tau$ (Fig. 4c) when fitting $I(t)$ in regime II with an exponential function $I(t) = I_0 \cdot \exp(-t/\tau)$ (Methods, Supplementary Fig. 12, Supplementary Note 6). Across extended regions of the sample, we find a strong correlation in the form $I_0 \cdot \tau \approx const$ (Fig. 4d). While such behavior is highly unexpected for a generic exponential decay, it emerges naturally from the form of $I(t)$ in Eq. 1.

Establishing the actual decay law of $I(t) \propto -\dot{P}$ reveals a more nuanced picture of relaxation dynamics of charge carriers $n(t) \propto P$ trapped at the domain wall. The behavior in Eq. 1 suggests that $n(t) \propto \log(t + \text{const})$. To understand this unexpected behavior, we notice that it is a solution of a modified kinetic equation, with a density-dependent relaxation rate $\gamma(n)$:

$$dn/dt = -\gamma(n)n \equiv -\gamma_0 \exp(n/\bar{n})n, \tag{2}$$

for $n(t) \gtrsim \bar{n}$. The exponential sensitivity of the tunneling rate through the potential barrier on $n$ is natural since the accumulation of electrons and holes on opposite sides of the domain walls leads to a decrease in the effective band gap and subsequent exponential growth of the tunneling rate (Fig. 4e). General consideration yield $\bar{n} \sim 0.1\,a^{-2}$, $a$ standing for perovskite lattice constant (Supplementary Note 10, ref. 43). The exponential sensitivity of tunneling to domain wall charging also explains the otherwise puzzling extreme sensitivity of the relaxation rate and magnitude of photocurrent to ambient light, as shown in Fig. 4a and Supplementary Fig. 13 (pink vs. blue curves). Indeed, weak intrinsic recombination rates imply that even small amounts of stray light can significantly modify photocurrent transients.

The spatial separation of electrons and holes at domain walls strongly suppresses recombination, exponentially enhancing the effective lifetimes and thereby allowing charge carriers to diffuse over large distances. In this sense, charged domain walls can serve as effective conductive pathways for electrons and holes (Fig. 4f), akin to the free electron gas at charged 90° domain walls in ferroelectric BaTiO$_3$[44] or the ferroelectric highways proposed by Frost et al. [13] in their study on the potential ferroelectric ordering in LHPs.

Flexoelectric domain walls can act as conductive channels for accumulated charges, enhancing charge transfer in the perovskite layer in a photovoltaic device. However, excessive charge accumulation could also increase tunneling rates and accelerate recombination of carriers trapped at domain walls, potentially limiting device efficiency under high illumination conditions. Understanding the extent of these competing effects and their impact on real-life device performance is therefore crucial for maximizing the efficiency of perovskite solar cells. This prompts a systematic investigation of flexoelectric domain walls in perovskite materials that will combine experimental characterization and theoretical modeling.

In summary, we present evidence of local inversion symmetry breaking in the nominally cubic high-temperature phase of MAPbBr$_3$, explicitly confirmed by a finite zero-bias two-photon bulk photovoltaic effect. We reconcile this finding with the previously established centrosymmetric structure of MAPbBr$_3$ by showing that the observed short-circuit photocurrent arises from local flexoelectric polarization at the boundaries between domains of spontaneous strain, which is present in as-grown bulk MAPbBr$_3$ even in the nominally cubic phase. We detect this strain through the resulting optical anisotropy and visualize the emerging domain structure using an original technique based on electrochemical staining of the domain walls. Our results further indicate that local electric fields induced by flexoelectricity lead to spatial separation of electrons and holes on the opposite sides of the domain boundaries. To recombine, the photocarriers must tunnel through the flexoelectric potential, which exponentially suppresses the recombination rate. As these long-lived charges can still move freely along the domain walls, the latter can serve as efficient mesoscale transport channels, enabling long-range diffusion essential for high photovoltaic efficiency. By revealing the microscopic origins of slow charge dynamics in bulk LHPs, we identify their mesoscopic structure as the origin of unique photoelectronic properties of these materials. Our findings unify seemingly contradictory observations into a coherent framework, providing a promising pathway for the design and optimization of hybrid perovskites in photovoltaic applications through mesoscopic structural engineering rather than compositional searches alone.

## Methods

### Materials
CH$_3$NH$_3$Br (>99.99%) was purchased from GreatCell Solar Ltd. (formerly Dyesol) and used as received. PbBr$_2$ (98%) and DMF (anhydrous, 99.8%) were purchased from Sigma Aldrich and used as received.

### Growth of CH$_3$NH$_3$PbBr$_3$ perovskite single crystals
A 1.5 M solution of CH$_3$NH$_3$Br/PbBr$_2$ in DMF was prepared, filtered through a 0.45 μm-pore-size PTFE filter, and the vial containing 0.5-1 ml of the solution was placed on a hot plate at 30 °C. The solution was then gradually heated to 60 °C and maintained at this temperature until the formation of CH$_3$NH$_3$PbBr$_3$ crystals. The crystals can be grown into larger sizes by elevating the temperature further. The crystals were collected and cleaned using a Kimwipe paper.

### Device fabrication
For domain wall visualization, wires were attached to the single crystal samples using 8330D silver conductive epoxy (MG Chemicals). The epoxy was cured at room temperature for 24 h. For photocurrent measurements, the single crystal sample was mounted on a glass holder, and the wires were attached to it using flexible carbon conductive epoxy G6E-FRP (Graphene Laboratories, Inc.). After curing at room temperature for 24 h, the epoxy adhesive provided both mechanical fixation and reliable electrical contact. The edges of the sample were covered with non-conductive epoxy to ensure isolation between adjacent contacts.

### Birefringence measurements
For the birefringence measurements, we used pristine, as-grown samples that had never been exposed to an electric field.

To visualize the natural birefringence in MAPbBr$_3$, the sample was put in a rotating crossed polarizer setup and illuminated with an expanded He-Ne laser beam (Thorlabs HNL050LB). The beam polarization was adjusted using a half-wave plate to match the orientation of the first polarizer (P1 in Fig. 1c). Transmission images of the sample were acquired at each position of the crossed polarizer using a CMOS camera (Allied Vision Alvium 1800 U-319m).

To study the critical behavior of birefringence across the cubic-to-tetragonal phase transition, a near-infrared ($\lambda = 1028$ nm) beam polarized at 45° relative to the [100] crystal axis was sent through the sample, placed inside an optical cryostat, normal to the (001) crystal face. At each temperature, a complete set of Stokes parameters was measured using the standard procedure[45]. Special care was taken to compensate for the thermal-expansion-driven displacement of the sample during the experiment (Supplementary Fig. 1, Supplementary Note 1).

### Electrochemical staining
For domain wall visualization, we used samples from the same batch as the sample used in the birefringence measurements shown in Fig. 1d, the photocurrent measurements presented in Fig. 3 and Fig. 4, and the X-ray diffraction measurements shown in Supplementary Fig. 6. Silver ions were generated in situ using silver epoxy electrodes attached to opposite sides of a crystal as the source of ions (Fig. 2a, Supplementary Fig. 2a). Oxidation of metallic silver at the electrode interface, ion transport into the bulk crystal, and subsequent electrochemical reduction back to metallic silver were all achieved in a single step by passing a small current through the sample. Experiments were performed in a light-tight box to suppress photocarrier generation, which would otherwise cause an undesirable increase in bulk conductivity.

Applied voltages ranged from 5 to 100 V mm$^{-1}$ with corresponding currents in the nA range depending on the sample. After several hours, dark-colored metallic silver structures appeared within the bulk crystal.

## Microscopy

Bright-field and confocal microscopy were performed using a Zeiss LSM 880 Confocal Laser Scanning inverted microscope equipped with a Zeiss Plan-Apochromat 10×, 0.45 NA objective and PMT detectors for both confocal and transmitted light imaging. A He-Ne laser with $\lambda = 632.8$ nm was used for illumination. The sample was placed in a Petri dish with a glass coverslip bottom. The refractive index matching liquid Immersol 518 F (Carl Zeiss Jena GmbH) was introduced between the sample and the coverslip to reduce reflection from the sample surface. Optical sectioning was performed with 5 μm resolution along the optical axis and lateral resolution of 1.66 μm per pixel. Illumination, focusing, optical sectioning, and primary image acquisition were controlled by Zeiss ZEN 2.3 SP1 Black software. Images were processed using Zeiss ZEN 2.3 SP1 Black, NIH ImageJ 1.54p, and Helicon Focus 8.3.0 software. Further information can be found in Supplementary Note 2.

## Photocurrent measurements

For the photocurrent measurements, we used pristine, as-grown samples that had never been exposed to an electric field.

The laser pulse source used was the Light Conversion Pharos HP, with a pulse energy of 2 mJ per pulse at a repetition rate of 3 kHz, a central wavelength of $\lambda = 1028$ nm, and a pulse duration of $\tau = 290$ fs. Only a small part of the full laser power was used for the experiment to avoid sample damage. The actual average pump laser power was $W = 0.25$ mW at a 1.5 kHz pulse repetition rate after the built-in laser pulse picker. The laser was focused using a lens with a focal distance of $F = 200$ mm, and the beam waist diameter inside the sample was $w = 40$ μm. To produce the spatially resolved photocurrent maps, the pump laser spot was scanned over the sample surface (Fig. 3b, c), with two pairs of carbon-based epoxy contacts attached to the sample, allowing for the detection of photocurrent flowing in both vertical and horizontal directions. The corresponding pair of contacts was connected directly to the current input of the lock-in amplifier (Zurich Instruments MFLI), leaving the other pair of contacts open. The measurements were performed using a lock-in amplifier in current mode (no bias) referenced to the laser output. The scanned area was limited to the part of the crystal away from the electrodes, thereby eliminating electrode proximity effects and excluding the possibility of charge carriers reaching the contacts by diffusion. The detected photocurrent was independent of the scan direction and pump polarization.

Time-resolved transients of photocurrent were reconstructed from the current frequency components acquired by lock-in at integer multiples of the laser repetition rate (up to $n = 102$ for high-resolution curves, such as in Fig. 3g and Fig. 4a, and $n = 18$ for area scans in Fig. 4b, c). The amplitude and phase of each harmonic of the photocurrent were measured separately, and then the time-domain representation of the signal was reconstructed by inverse Fourier transform of the frequency domain data

$$I(t) = \sum_{k=1}^{n} \left[ X_k \cos(2\pi f_k t) - Y_k \sin(2\pi f_k t) \right], \quad (3)$$

where $X_k$ and $Y_k$ are the real and imaginary parts of the $k$-th harmonic $f_k$ of the laser repetition rate, respectively. For the purpose of time-domain analysis of the current transients, the DC component appearing in the reconstructed signal is subtracted. Further information can be found in Supplementary Note 5 and 6 and the accompanying figures.

## Data availability

The relevant data supporting the findings of this study are available within the Article, the Supplementary Information file, and the Source Data file. All raw data generated in this study are available from the corresponding author upon request. Source data are provided with this paper.

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

## Acknowledgements

We are grateful to A. G. Volosniev for the valuable discussions. We thank D. Milius for the assistance with microscopy. D. R. would like to thank F. Filakovský and T. Čuchráč for the valuable discussions. This research was supported by the Scientific Service Units (SSU) of ISTA through resources provided by the Imaging & Optics Facility (IOF) and the Miba Machine Shop Facility (MS).

## Author contributions

D.R. and Z.A. conceptualized the project, designed the experiments, and analyzed the data. A.A. Z. and O.M. B. synthesized the samples. D.M.B. measured and analyzed the X-ray data, D.L. and Z.A. measured the temperature dependence of birefringence, and D.R. conducted the rest of the experiments. The manuscript was written by D.R. and Z.A., with feedback from all coauthors. Z.A. supervised the project.

## Competing interests

The authors declare no competing interests.
