## [Transparent Peer Review file · Nature Communications]

Flexoelectric domain walls enable charge separation and transport in cubic perovskites

Corresponding Author: Dr Zhanybek Alpichshev

Version 0:

Reviewer comments:

Reviewer #1

(Remarks to the Author)

This work investigates flexoelectric polarization in MAPbBr₃ single crystals through a combination of optical and charge transport measurements. The experimental approach is distinct compared to previous studies on ferroelectric and ferroelastic behavior in similar material systems. However, it remains unclear how the presented measurements provide direct evidence for local inversion symmetry breaking. In particular, the use of polarized optical microscopy to observe strain and domain structures is insufficient to resolve symmetry breaking at the micro- to nanoscale. The authors state that “the sign of photocurrent I_{rep} is then determined by the average slope of electrostatic potential,” but this assertion does not constitute direct evidence of an electrostatic potential distribution within the sample. Overall, the measurements presented in Figures 3 and 4 do not conclusively demonstrate the presence of microscopic polarization in MAPbBr₃. Therefore, I do not recommend this manuscript for publication in Nature Communications.

1. Figure 2 shows domain walls in MAPbBr₃ after applying an electric field of 24 V/mm for 4 hours. However, no data are provided on the evolution at shorter timescales (e.g., 10 minutes, 1 hour) or whether the domain wall patterns stabilize after a certain duration. Please provide time-dependent microscopy images and clarify whether the silver dendrite formation reaches a steady state or continues to evolve beyond 4 hours.
2. Please provide the absorption or diffuse reflectance spectrum of the crystal and briefly explain why a 632.8 nm laser was chosen for the birefringence measurement.
3. The descriptions of bright-field and confocal microscopy (Materials and Methods, Supplementary Text S2) are insufficient for readers unfamiliar with these techniques. Please provide a brief explanation.
4. Figure 1e shows that birefringence decreases above T_c . Can the authors confirm whether the sample appears dark when placed between crossed polarizers above T_c ?
5. In lines 150-153, the authors claim that the silver structures start to dissolve after several hours. But there is no experimental evidence to support the statement. Please provide the data.
6. Please explain why a 1028 nm NIR laser was chosen to study the photochromic effect. What is the behavior under 632.8 nm laser excitation?
7. The photochromic effect is observed in silver-doped MAPbBr₃, but no reference data for undoped MAPbBr₃ are provided. Does the undoped crystal exhibit any photochromic behavior under similar conditions?
8. The manuscript attributes birefringence to bulk strain and ferroelastic domain boundaries (lines 124–126). However, the spatial patterns observed in cross-polarized microscopy (Figure 1d) differ from those in electrochemical staining (Figure 2). Why do these techniques reveal different patterns? Please clarify the relationship between bulk strain and domain boundaries.
9. The potential influence of ionic migration is not quantitatively discussed. Could the structural defects or ferroelastic domains be induced by ion migration under an electric field or light?
10. The unit of the color bar in Figure 3c needs to be clarified.
11. How long does it take to complete one position-dependent photocurrent scan? Does the scan direction affect the result?
12. The manuscript claims that photocurrent in MAPbBr₃ is generated via two-photon absorption using a 1.2 eV laser. Please provide experimental evidence, such as power-dependent photocurrent or photoluminescence measurements.

Reviewer #2

(Remarks to the Author)

Reviewer #3

(Remarks to the Author)

The authors report that the carrier lifetimes and diffusion lengths in nominally cubic MAPbBr₃ arise from flexoelectric polarization confined to microscopic ferroelastic domain walls. Using polarized-light microscopy and electrochemical silver-staining, they show that strain gradients at domain interfaces create local electric fields which spatially separate and trap charge carriers, dramatically suppressing recombination and enabling long-lived photocurrents under zero bias. There is a simple but powerful story based on innovative experiments. I believe this work will generate significant interest in the community, and I recommend acceptance subject to minor revisions.

Suggestions

- Mention the composition studied in the abstract. It could also benefit from some more detail.
- Figure 1e fit details: Include goodness-of-fit (R^2) to justify the extracted critical exponent.
- Laser-spot characterization: Confirm that the 40 μm waist did not induce nonlinear damage.
- Ambient-light sensitivity: Comment on implications for device operation under real-world lighting.
- Further confirmation of local symmetry breaking has been recently reported in Nature Nanotechnology 20, 755 (2025).
- Generality beyond MAPbBr₃: Mention whether similar flexoelectric domain structures are expected in iodide perovskites (e.g., MAPbI₃) or mixed-halide systems.
- The "CH₃NH₃Br₃" precursor and "Growth of CH₃NH₃PbBr" in the methods section.

Reviewer #4

(Remarks to the Author)

The manuscript proposes a novel mechanism for achieving charge separation and transport in perovskites via flexoelectric domain walls, which is highly interesting and innovative. To enhance the persuasiveness of the manuscript, the following suggestions are provided.

(1) The authors propose that the observed phenomena originate from the flexoelectric effect, yet they have not provided substantive evidence. Further in-depth exploration is needed regarding the specific action mechanism of the flexoelectric effect at the microscopic scale, such as detailed simulations or experimental verifications of the electric field distribution at domain walls.

(2) The manuscript employs an electrochemical staining technique to visualize the domain wall structures in MAPbBr₃, which is a novel and effective approach. However, the description of the technical details is relatively brief. It is recommended to provide a detailed account of the specific parameters used in the staining process, such as the magnitude of the current, the duration of treatment, and the temperature conditions.

(3) As mentioned in the previous comment, given the potential complexities introduced by electrochemical staining (e.g., the diffusion and reduction processes of silver ions may affect the true morphology of domain walls), it is advisable to explore alternative non-destructive or low-damage techniques for visualizing domain walls. Techniques such as Piezoresponse Force Microscopy (PFM) could be employed to provide more direct and accurate information on domain wall structures.

(4) The manuscript primarily focuses on the charge behavior at domain walls; however, since these domain walls originate from defects, which may involve defect recombination, the impact of domain wall dynamics on the overall stability of the material has not been addressed. It is recommended to investigate how the formation and annihilation states of domain walls influence the long-term stability and photoelectric conversion efficiency of the material.

(5) The current photocurrent measurements reflect the overall response between electrodes, which do not prove that the photo-generated carriers indeed undergo long-distance transport along specific domain wall pathways rather than through the bulk phase. More direct spatially resolved evidence is needed. It is recommended to attempt utilizing microscopic photocurrent imaging techniques (such as Laser Beam Induced Current Imaging - LBIC, or Scanning Photocurrent Microscopy - SPCM) to intuitively map the correlation between photocurrent generation hotspots and domain wall positions at the micrometer/sub-micrometer scale, and observe whether the current truly flows along the domain walls.

(6) In real polycrystalline perovskite thin films, in addition to domain walls, grain boundaries, surfaces, defect states, etc., may all serve as pathways or traps for carrier transport. The manuscript needs to delve more deeply into the competitive relationship between the discovered "flexoelectric domain wall channels" and the intrinsic transport pathways, which is crucial for evaluating their practical device application value.

(7) The manuscript proposes the potential application of flexoelectric domain walls in perovskite solar cells, yet it does not thoroughly explore the commercialization pathways and associated challenges. It is recommended to add a dedicated section discussing how this discovery can drive further advancement in perovskite solar cell technology, including an analysis of potential market barriers, scalability issues, cost-effectiveness considerations, and strategies for overcoming

technical hurdles in the transition from laboratory research to commercial products.

Version 1:

Reviewer comments:

Reviewer #1

(Remarks to the Author)

While the authors have addressed several reviewer questions, some points remain unclear in the current manuscript. First, why did the authors choose MAPbBr₃ instead of CsPbBr₃, which adopts a perfectly cubic structure at room temperature? Could the authors provide similar experimental results on CsPbBr₃ crystals? This comparison would help clarify whether the proposed method for understanding the flexoelectric effect in perovskites is general or specific to the organic–inorganic hybrid system.

Furthermore, since MAPbBr₃ is not a representative material for state-of-the-art photovoltaic devices, the authors should explain how the present study offers new insights or fundamental understanding relevant to more practical photovoltaic compositions, such as mixed-cation lead iodide perovskites.

Regarding Ag doping in MAPbBr₃, what role does Br[−] ion migration play in defect formation induced by Ag incorporation? Could such halide ion migration contribute to the observed flexoelectric effect and local symmetry breaking?

Lastly, in response to Reviewer 4, the authors mention that the preservation of inversion symmetry in bulk perovskites has been confirmed by optical SHG experiments (Refs. 37 and 38). However, upon reviewing these references, the materials studied are not MAPbBr₃. I do not believe that MAPbBr₃ exhibits any SHG response, as its centrosymmetric structure should preclude second-order nonlinear optical activity. To my knowledge, no SHG signal has been reported for this material.

Reviewer #2

(Remarks to the Author)

Reviewer #4

(Remarks to the Author)

The author has answered the relevant questions. The current version meets the requirements of the journal. I have no further doubts now.

Version 2:

Reviewer comments:

Reviewer #1

(Remarks to the Author)

I do not have further comments or questions on this manuscript.

Reviewer #2

(Remarks to the Author)

REVIEWER COMMENTS

Reviewer #1 (Remarks to the Author):

This work investigates flexoelectric polarization in MAPbBr₃ single crystals through a combination of optical and charge transport measurements. The experimental approach is distinct compared to previous studies on ferroelectric and ferroelastic behavior in similar material systems. However, it remains unclear how the presented measurements provide direct evidence for local inversion symmetry breaking.

We thank the Referee for taking the time to review our work. Coming to the concern whether our data indicate the breaking of inversion symmetry in room-temperature MAPbBr₃, we respectfully disagree with the Referee. As will be elaborated on below, there are multiple experimental observations that evidence the lack of central symmetry, among which we believe the observation of zero-bias photocurrent is the most straightforward. Photocurrent consists of the directional flow of charged photocarriers. The fact that this flow has a well-defined direction in the absence of external bias in a nominally cubic material indicates the lack of inversion symmetry within the volume of photo-generation. Alternatively, electric current is classified as inversion-odd (polar) vector. It has to be identically equal to zero in a centrosymmetric medium in the absence of explicit breaking of inversion symmetry by external means (e.g., electric bias). Therefore, the observation of photocurrent generation in the absence of bias constitutes direct evidence for the breaking of inversion symmetry in the nominally cubic high-temperature phase of MAPbBr₃. Since the bulk polar phase in MAPbBr₃ is not consistent with the existing empirical evidence (e.g., the lack of second-harmonic generation in MAPbBr₃), we conclude that the inversion symmetry breaking leading to the observed zero-bias photocurrent must be of a localized nature, rather than reflecting the bulk symmetry of the material.

In particular, the use of polarized optical microscopy to observe strain and domain structures is insufficient to resolve symmetry breaking at the micro- to nanoscale.

We agree with the Referee that polarized optical microscopy is insufficient to differentiate between polar and non-polar (nematic) pathways of cubic symmetry breaking. However, in this specific case, the results of microscopy indicate that the local refractive index tensor $n_{\alpha\beta}$ is not only birefringent but also non-uniform in space. This, in turn, implies that its gradient is nonzero. Even though $n_{\alpha\beta}$ by itself is indeed an inversion-even object, its gradient $\nabla_{\alpha}n_{\beta\gamma} \neq 0$ manifestly breaks inversion symmetry.

Beyond this general consideration, our main evidence for the absence of inversion symmetry is the observation of zero-bias photocurrent, which – as we argue above – cannot be non-zero in a centrosymmetric setting. Once we establish the fact that inversion symmetry in nominally cubic MAPbBr₃ is compromised, we proceed to determine the microscopic origin of this symmetry lowering.

The authors state that “the sign of photocurrent I_{rep} is then determined by the average slope of electrostatic potential,” but this assertion does not constitute direct evidence of an electrostatic potential distribution within the sample.

We thank the Referee for pointing out this potential source of confusion. However, the purpose of this argument is not to assert a specific spatial profile for the electrostatic potential, but to resolve the apparent contradiction between the observation of zero-bias photocurrent indicating broken inversion symmetry and the previously reported centrosymmetric nature of the bulk of MAPbBr₃.

The additional complication of accepting the existence of a nonzero gradient of electrostatic potential in extended volumes is that it is not obvious how to reconcile it with the reported high ionic conductivity of lead halide perovskites. Indeed, the mobile ionic carriers would tend to regroup within the material under the effect of resultant electric fields, eventually screening all potential gradients to zero.

As we argue in the paper, the seeming paradox is naturally resolved if non-centrosymmetry is localized to a (ideally) zero-measure subset of the total volume of the sample. The polarized microscopy indicates that this subset corresponds to the interfaces between centrosymmetric domains. Due to the displacement nature of the observed zero-bias photocurrent, the distribution of locally excited photocurrent magnitudes provides a direct mapping of the local electrostatic potential distribution averaged over the illuminated volume of the sample in vertical or horizontal direction (Fig. 3c in the original manuscript). Without the applied bias, the only possible source of the detected photocurrent is the local electrostatic potential.

Overall, the measurements presented in Figures 3 and 4 do not conclusively demonstrate the presence of microscopic polarization in MAPbBr₃.

We respectfully disagree with the Referee here. As argued above, the polar nature of (photo)current vector necessitates breaking inversion symmetry for it to be non-zero. Therefore, the observation of the zero-bias photocurrent generation under local illumination provides direct evidence of polar order in MAPbBr₃. The microscopic origin of inversion symmetry breaking is later identified through a combination of extensive experimental studies using various techniques along with prior knowledge, which offers a comprehensive understanding of the source of the photocurrent found to be driven by the flexoelectric effect at domain walls.

Therefore, I do not recommend this manuscript for publication in Nature Communications.

We thank the Referee for reading the manuscript and providing valuable comments. We believe that we thoroughly addressed the questions raised by him/her. We further reinforced the key points of our paper with new data and believe it is now suitable for publication in *Nature Communications*.

1. Figure 2 shows domain walls in MAPbBr₃ after applying an electric field of 24 V/mm for 4 hours. However, no data are provided on the evolution at shorter timescales (e.g., 10 minutes, 1 hour) or whether the domain wall patterns stabilize after a certain duration. Please provide time-

dependent microscopy images and clarify whether the silver dendrite formation reaches a steady state or continues to evolve beyond 4 hours.

To address the Referee's concern regarding the stability of domain wall patterns revealed through the formation of silver dendrites, we carried out the suggested time-dependent measurements. Microscopic images taken before (“0 minutes”) and at 60, 90, 120, 180, and 240 minutes after the current began to flow through the sample were added to Section S2 of the revised Supplementary Information, along with a corresponding description. Supplementary Fig. S3 shows that the staining process occurs gradually: initial silver (dendritic) structures appear after approximately 90 minutes; additional dendrites form progressively during subsequent applications of voltage across the sample. Importantly, the structures revealed at each step remain unchanged in all later measurements, even as new dendrites continue to form in previously unstained regions. The domain wall patterns that were made visible through staining at 90, 120, and 180 minutes can therefore be considered stable. No attempt was made to track the evolution of the structures beyond 4 hours due to the high risk that silver dendrites would reach the opposite side of the crystal, connect the electrodes directly, and cause a short circuit, which could destroy the sample and potentially damage the equipment.

As noted in the manuscript (lines 150-153 in the original manuscript), the silver dendrites begin to fade away once the electric field is removed, indicating these are not stable (equilibrium) structures. The microscope used for tracing the growth of dendrite structure does not allow for the *in situ* application of voltage inside the microscope. To cope with that, we had to subject the sample to electric bias for a given duration, perform imaging, and reapply bias until the next imaging round. Therefore, it should be considered that since microscopic imaging is a time-consuming process that can take several hours, structures formed through a step-by-step process may not be identical to those produced by a single-step procedure. Recurrent application of voltage with sufficiently long off-times, during which the already-formed structures can partially dissolve, can have unpredictable effects on the final structures. However, studying the effects of intermittent dissolution on the growth of silver dendrites within single crystals of MAPbBr₃ falls beyond the scope of this study.

2. Please provide the absorption or diffuse reflectance spectrum of the crystal and briefly explain why a 632.8 nm laser was chosen for the birefringence measurement.

In choosing the specific value for the probe wavelength, we relied on the optical and near-infrared absorption spectra of lead-halide perovskites, well-documented in literature (for example, ref. 3 for MAPbBr₃). Since the band gaps of lead-halide perovskites are famously free of deep trap states, any radiation with photon energy below the absorption threshold (corresponding to bandgap energy ($\Delta_g = \sim 2.3$ eV for MAPbBr₃) and away from the absorption lines of the MA cation (the highest one at ~ 0.3 eV; see e.g., doi: 10.1021/acs.jpcc.5b07432 and doi: 10.1021/acs.jpcclett.3c01158) will fall into the transparency range of MAPbBr₃ and can be used to examine its birefringence. In this work, the birefringence measurements were performed using 632.8 nm (1.96 eV) (Fig. 1d) and 1028 nm (1.2 eV) (Fig. 1e and Supplementary Fig. S8c) wavelengths. For imaging the entire crystal, the output of a continuous wave (CW) He-Ne laser at 632.8 nm was chosen for convenience (ease of alignment and detection).

3. The descriptions of bright-field and confocal microscopy (Materials and Methods, Supplementary Text S2) are insufficient for readers unfamiliar with these techniques. Please provide a brief explanation.

We thank the Referee for pointing this out. A description of the bright-field and confocal microscopy was added to Section S2 of the revised Supplementary Information.

4. Figure 1e shows that birefringence decreases above T_c . Can the authors confirm whether the sample appears dark when placed between crossed polarizers above T_c ?

We thank the Referee for bringing this question up. This is indeed an important matter, key to the entire narrative. As explained in the manuscript (lines 110-112 in the original manuscript) and Supplementary Information (Supplementary Text S1 in the original Supplementary Materials), the sample does not become completely dark above critical temperature T_c when placed between crossed polarizers, as shown in Fig. 1d. This implies that MAPbBr₃ is birefringent even in the nominally cubic phase, which in turn implies that the structure is in fact not cubic there. What we plot in Fig. 1e is the *change* of birefringence across the phase transition at T_c , with residual high-temperature birefringence acting as a baseline (marked as zero-level in Fig. 1e). As described in the manuscript, the key message of the data in Fig. 1e is that even though the birefringence, that acts as a measure of non-cubicity, is already nonzero in the high-temperature phase, the transition at T_c remains sharp. This implies that the nature of non-cubicity above T_c is distinct from the tetragonal distortion that sets in below T_c .

5. In lines 150-153, the authors claim that the silver structures start to dissolve after several hours. But there is no experimental evidence to support the statement. Please provide the data.

We agree that in the submitted version of the manuscript, this information was simply stated verbally as a description of our observations. Following the advice of the Referee, we conducted additional measurements, documenting the dissolution process this time. The microscopic images showing the gradual dissolution and the nearly-complete disappearance of silver dendrites in MAPbBr₃ over several days after the turn off of the bias (Supplementary Fig. S4) have been added to Section S2 of the revised Supplementary Information, along with a corresponding description.

6. Please explain why a 1028 nm NIR laser was chosen to study the photochromic effect. What is the behavior under 632.8 nm laser excitation?

We thank the Referee for bringing up this question. The short answer is that we only observed photochromism in MAPbBr₃ when using pulsed radiation, while continuous-wave radiation at 632.8 nm (He-Ne laser) did not produce any detectable effect. We believe that the specific value of the wavelength being 1028 nm is not important; the actually relevant parameter is the peak intensity, which only achieves high values in pulsed laser sources.

Before expounding on the possible reason for such behavior, we would like to state here that the photochromism of silver-doped MAPbBr₃ is a newly discovered phenomenon, which is only tangentially related to the main message of the present work and has to be investigated as separate project to reach full understanding.

At present, the body of existing preliminary evidence indicates that photochromism in silver-doped MAPbBr₃ is associated with the reversible photochemical reduction of Ag⁺ ions electrochemically injected into the perovskite matrix. This implies that the mechanism of the photochromic effect is likely similar to that observed in silver-halide-sensitized silicate glasses. In the latter, the silver-halide nanocrystals are embedded in the glass matrix, and the reduction of Ag⁺ ions occurs by capturing photoelectrons ejected from the halogen ion by incident UV or visible light, resulting in the formation of small aggregates of silver atoms absorbing light. Since the reaction products remain in the reaction volume, the process is completely reversible, and silver nanoclusters dissolve when the glass is heated or left unperturbed in darkness at room temperature.

Guided by this understanding, we conjecture that the main mechanism of bulk photochromic effect in silver-doped MAPbBr₃ occurs through either linear (if the photon energy exceeds the bandgap) or multi-photon (if it doesn't) absorption, and subsequent capture of the photoelectrons by Ag⁺ ions. The bandgaps of both AgPbBr₃ (~ 4.0 eV, ~ 310 nm) and AgBr (~ 2.5 eV, ~ 496 nm), which can potentially be present in the form of nanocrystals within the silver-doped MAPbBr₃ matrix, exceed the bandgap of MAPbBr₃ (~ 2.3 eV, ~ 535 nm). Therefore, in the bulk of silver-doped MAPbBr₃, the photoelectrons necessary for reducing Ag⁺ ions can only be produced through a multi-photon process. This naturally explains why, when using 632.8 nm (1.96 eV) or 1028 nm (1.2 eV) sub-gap radiation, we only see clear photochromism with ultrashort 1028 nm pulses (peak intensity $I_{\text{peak}} \sim 10^9 \text{ W/cm}^2$) and not with continuous-wave output of He-Ne laser at 632.8 nm with intensity $I_{\text{CW}} \sim 10^2 \text{ W/cm}^2$.

In conclusion, we would like to reiterate that while the photochromism of silver-doped MAPbBr₃ is indeed a very interesting phenomenon that deserves a separate dedicated research project, which would include investigating how silver-doped samples respond to different wavelengths and intensities of light. However, the focus of the present study is the structure and optoelectronic properties of undoped pristine MAPbBr₃, with a detailed study of the photochromic effect falling outside its immediate scope.

7. The photochromic effect is observed in silver-doped MAPbBr₃, but no reference data for undoped MAPbBr₃ are provided. Does the undoped crystal exhibit any photochromic behavior under similar conditions?

This is a valid concern, in light of the novelty of the observed effect. The answer is that we do not observe the photochromic effect in undoped MAPbBr₃ samples, and to our knowledge, there are no reports of this effect in MAPbBr₃ in the literature. As mentioned in the manuscript (lines 153-156 in the original manuscript) and elaborated in Section S3 of the Supplementary Information (Supplementary Text S3 of the original Supplementary Materials), the observed photochromism is associated with the photochemical reduction of Ag⁺ ions, a process well-known for causing the photochromic activity of silver-halide-sensitized silicate glasses and widely used in conventional photography.

Motivated by this comment, we performed reference measurements on undoped MAPbBr₃ under conditions similar to those we used to observe photochromicity in silver-doped samples. We

confirmed that the effect is absent in undoped samples, as shown in Supplementary Fig. S5 of the revised Supplementary Information.

8. The manuscript attributes birefringence to bulk strain and ferroelastic domain boundaries (lines 124–126). However, the spatial patterns observed in cross-polarized microscopy (Figure 1d) differ from those in electrochemical staining (Figure 2). Why do these techniques reveal different patterns? Please clarify the relationship between bulk strain and domain boundaries.

We thank the Referee for raising this valid concern. To respond, we would like to begin with some technical details of the experimental methods used.

The sample used in the birefringence measurements shown in Fig. 1d was later utilized in the photocurrent measurements presented in Fig. 3 and Fig. 4 (the photo of the same sample can be seen in the inset of Fig. 4d), as well as in X-ray diffraction measurements shown in Supplementary Fig. S6. To prevent silver ions from diffusing into the material and potentially affecting the measurement results, carbon-based epoxy contacts were attached to the crystal for the photocurrent measurements. Although this information is provided in Sections S4 and S5 of the Supplementary Information (Supplementary Texts S4 and S5 of the original Supplementary Materials), we included it in the main text of the revised manuscript to avoid possible confusion. For domain walls visualization, different samples from the same batch had to be used, as the process of electrophoretic silver-doping is apparently non-reversible.

Coming back to the actual question, generally speaking, there is no reason to assume that the birefringence map would match the pattern of domain boundaries. The birefringence maps presented, e.g., in Fig. 1d, were obtained in transmission through bulk sample. This means that what we see here at every point is the cumulative birefringence integrated over the entire thickness of the sample (~ 2 mm). With a typical domain size on the order of μm , the measured birefringence is averaged over hundreds or even thousands of domains, each contributing to the phase shift acquired by the beam passing through the sample. As explained in the manuscript, this is why we observe smooth gradients of birefringence without corresponding gradients of stress (lines 157–161 in the original manuscript). The purpose of measuring birefringence in the nominally cubic high-temperature phase of MAPbBr_3 was to demonstrate that the cubic symmetry is broken and that non-uniform strain exists in a typical free-standing (without external stress applied) single crystal. Other techniques, including electrochemical staining of domain walls, are used to clarify how exactly the symmetry is broken.

9. The potential influence of ionic migration is not quantitatively discussed. Could the structural defects or ferroelastic domains be induced by ion migration under an electric field or light?

We appreciate the Referee pointing out the potential role of ionic migration. Indeed, ion migration under an electric field or illumination is a well-documented phenomenon in lead-halide perovskites. However, in our study, no electric field was applied to the samples used for optical, photocurrent, and X-ray measurements, indicating that the polar order associated with flexoelectric domain walls is an intrinsic property of typical as-grown single crystals samples of MAPbBr_3 . In fact, as mentioned above, the concern for mobility of ions in perovskites is one of the arguments for the inversion symmetry breaking to be localized to domain wall boundaries. Indeed, if one

were to assume there is a static electric field across macroscopic distances within domains, it would be hard to explain why that field does not give rise to long-range transport of ions that would eventually screen away the field.

Coming to the non-equilibrium behavior, the possibility that light causes structural changes can be ruled out based on birefringence measurements performed with low-intensity CW 632.8 nm He-Ne laser (Fig. 1d) and a high-intensity pulsed 1028 nm laser (Supplementary Fig. S6c). Despite the orders-of-magnitude difference in peak intensity, the cross-polarized images of the sample are nearly identical for both light sources. Additionally, the 1028 nm light was used to repeatedly measure birefringence on the same sample with negligible differences between individual measurements.

Regarding the electrochemical staining experiments, additional time-dependent measurements were carried out to assess the stability of the revealed domain wall patterns (Supplementary Fig. S3 and Section S2 of the revised Supplementary Information). These measurements showed that the structures formed at the onset of the staining process remained unchanged despite repeated application of the electric field. This indicates that ionic diffusion under the low electric fields used in this work (10–100 V/cm) does not affect the intrinsic domain structure of MAPbBr₃.

10. The unit of the color bar in Figure 3c needs to be clarified.

We thank the Referee for noting this. The colorbar in Fig. 3c (Fig. 3e in the revised manuscript) shows the absolute scale for the related colormaps, with units indicated below the corresponding plots. To avoid confusion, the units have been moved to the right side of the colorbar.

11. How long does it take to complete one position-dependent photocurrent scan? Does the scan direction affect the result?

It takes about 10 hours (621 minutes) to complete a single position-dependent photocurrent scan (with 10 seconds spent at each point). The results are unaffected by scan direction and were consistently reproduced in repeated experiments. The photocurrent generated at each spot can be reproducibly measured independently (without scanning the entire crystal), as was observed in the case of multi-harmonic measurements at specific locations for reconstructing current pulse profiles (Supplementary Fig. S13 and Section S6 of the revised Supplementary Information). The magnitudes of I_{rep} obtained during these measurements are in good agreement with the values obtained during area scans, considering the experimental error. This additional information has been included in Section S5 of the revised Supplementary Information.

12. The manuscript claims that photocurrent in MAPbBr₃ is generated via two-photon absorption using a 1.2 eV laser. Please provide experimental evidence, such as power-dependent photocurrent or photoluminescence measurements.

The photocurrent is generated due to charge separation occurring at flexoelectric domain boundaries after the sample is excited with below-the-bandgap 1028 nm (1.2 eV) laser light. Observed photoluminescence (Supplementary Fig. S8d) indicates the generation of real photoexcited carriers via nonlinear absorption. The proximity of the energy of two 1.2 eV photons to the bandgap energy of the MAPbBr₃ ($\Delta_g = \sim 2.3$ eV) renders two-photon absorption the most

likely nonlinear process causing the excitation. We would like to refer the Referee to the recent review article on two-photon absorption in lead halide perovskites, doi: 10.1039/D1MH02074A, that summarizes a large body of literature on the topic, including studies performed on single-crystal MAPbBr₃. A detailed study of nonlinear absorption in MAPbBr₃ can also be found in a recent paper, doi: 10.1021/acsp Photonics.5c01360. The referenced papers are now included in the list of citations and cited in the revised manuscript (ref. 34, 35).

Following the advice of the Referee, we also conducted additional power-dependent measurements of photoluminescence, indicating two-photon absorption (please see new Supplementary Fig. S10). This has been added to Section S5 of the revised Supplementary Information, along with a corresponding description.

Reviewer #2 (Remarks to the Author):

We thank both Referees #1 and #2 for detailed constructive criticism. This has helped to significantly to improve the presentation quality of our work.

Reviewer #3 (Remarks to the Author):

The authors report that the carrier lifetimes and diffusion lengths in nominally cubic MAPbBr₃ arise from flexoelectric polarization confined to microscopic ferroelastic domain walls. Using polarized-light microscopy and electrochemical silver-staining, they show that strain gradients at domain interfaces create local electric fields which spatially separate and trap charge carriers, dramatically suppressing recombination and enabling long-lived photocurrents under zero bias. There is a simple but powerful story based on innovative experiments. I believe this work will generate significant interest in the community, and I recommend acceptance subject to minor revisions.

We appreciate the Referee's valuable feedback. The suggested changes have been incorporated into the revised manuscript as described below.

Suggestions

- Mention the composition studied in the abstract. It could also benefit from some more detail.

We thank the Referee for bringing this to our attention. We have revised the abstract to include their suggestions.

- Figure 1e fit details: Include goodness-of-fit (R^2) to justify the extracted critical exponent.

The goodness-of-fit (R^2) has now been included in the Fig. 1 caption of the revised manuscript.

- Laser-spot characterization: Confirm that the 40 μm waist did not induce nonlinear damage.

To demonstrate that the 1028 nm laser beam with a 40 μm beam waist used in our study did not cause nonlinear damage, we carried out additional time-resolved measurements of transmission and photoluminescence on a MAPbBr_3 single crystal sample (Supplementary Fig. S9 is added to Section S5 of the revised Supplementary Information). No significant changes in transmission or photoluminescence were observed throughout the course of one hour of continuous irradiation. Since in the actual experiment, measuring a single harmonic of photocurrent takes only 10 seconds, and about 6 minutes are spent at every spot to reconstruct the time-dependent profile of the current pulse at several selected points (by measuring the first 102 harmonics), it is unlikely that the beam causes any nonlinear damage within this timeframe.

- Ambient-light sensitivity: Comment on implications for device operation under real-world lighting.

Charge accumulation at flexoelectric domain walls may facilitate charge extraction from the perovskite layer but could also enhance tunneling and promote recombination at domain walls, potentially limiting device efficiency under high illumination. We have expanded the Discussion section in the revised manuscript to address these implications for the performance of real-world photovoltaic devices.

- Further confirmation of local symmetry breaking has been recently reported in Nature Nanotechnology 20, 755 (2025).

We thank the Referee for bringing this recent work up. It has been included in the reference list and cited in the Introduction section of the revised manuscript.

- Generality beyond MAPbBr_3 : Mention whether similar flexoelectric domain structures are expected in iodide perovskites (e.g., MAPbI_3) or mixed-halide systems.

We thank the Referee for raising this important point. We carried out additional measurements and discovered the two-photon bulk photovoltaic effect in MAPbI_3 by detecting the zero-bias photocurrent following localized optical excitation of a single crystal sample (Supplementary Fig. S14, Section S7 of the revised Supplementary Information). The implications of this finding are discussed in the Results section of the revised manuscript.

- The “ $\text{CH}_3\text{NH}_3\text{Br}_3$ ” precursor and "Growth of $\text{CH}_3\text{NH}_3\text{PbBr}$ " in the methods section.

We thank the Referee for noticing these errors. They have been corrected in the revised manuscript.

Reviewer #4 (Remarks to the Author):

The manuscript proposes a novel mechanism for achieving charge separation and transport in perovskites via flexoelectric domain walls, which is highly interesting and innovative. To enhance the persuasiveness of the manuscript, the following suggestions are provided.

(1) The authors propose that the observed phenomena originate from the flexoelectric effect, yet they have not provided substantive evidence. Further in-depth exploration is needed regarding the specific action mechanism of the flexoelectric effect at the microscopic scale, such as detailed simulations or experimental verifications of the electric field distribution at domain walls.

We thank the Referee for this valuable suggestion. We agree that a complete microscopic understanding of the flexoelectric effect would offer deeper insight. A systematic investigation combining detailed theoretical simulations with direct experimental mapping of local electric fields at domain walls is an important next step, but it lies beyond the scope of the present study. In this work, our conclusion regarding the role of the flexoelectric effect is based on the following evidence:

(1) explicit proof of local inversion-symmetry breaking in the nominally cubic high-temperature phase of MAPbBr₃, suggested by the presence of non-zero gradients of birefringence and confirmed by the observation of the zero-bias two-photon bulk photovoltaic effect;

(2) the displacement nature of zero-bias photocurrent detected after localized charge injection, consistent with the presence of internal electric fields driving charge separation. This was further confirmed by the additional data on the net charge transfer in a single current pulse, obtained by means of a purely analog electrometer; please refer to Supplementary Fig. S15 and Section S8 in the revised Supplementary Information;

(3) preservation of inversion symmetry in the bulk, as previously confirmed by optical second-harmonic generation experiments (ref. 37, 38);

(4) observation of non-uniform strain in nominally cubic MAPbBr₃ single crystals, implying the presence of defects such as domain walls;

(5) direct visualization of domain walls in electrochemical staining experiments; and

(6) the fact that domain walls are the only regions where strain gradients can generate local electric fields without breaking inversion symmetry in the bulk.

Collectively, these results strongly support localized flexoelectricity in nominally cubic MAPbBr₃ as the unifying mechanism that reconciles otherwise conflicting experimental observations, including both earlier reports and the findings presented here. We have revised the manuscript to state the evidential basis of our interpretation more clearly and to highlight future work directions.

(2) The manuscript employs an electrochemical staining technique to visualize the domain wall structures in MAPbBr₃, which is a novel and effective approach. However, the description of the technical details is relatively brief. It is recommended to provide a detailed account of the specific

parameters used in the staining process, such as the magnitude of the current, the duration of treatment, and the temperature conditions.

We agree with the Referee that the information provided in the Materials and Methods section of the original manuscript is insufficient. A full description of the procedure has been added to the Methods section of the revised manuscript.

(3) As mentioned in the previous comment, given the potential complexities introduced by electrochemical staining (e.g., the diffusion and reduction processes of silver ions may affect the true morphology of domain walls), it is advisable to explore alternative non-destructive or low-damage techniques for visualizing domain walls. Techniques such as Piezoresponse Force Microscopy (PFM) could be employed to provide more direct and accurate information on domain wall structures.

We thank the Referee for raising this important point. We agree that electrochemical staining can, in principle, introduce additional complexity through ion diffusion and reduction processes, and that care must be taken when interpreting the resulting domain-wall morphologies. In our study, the Ag-staining method was chosen because it offers high-contrast visualization over considerable volumes within the bulk, which was especially useful for the scope of this work since we are primarily interested in the intrinsic structure of MAPbBr_3 , with minimal extrinsic influence from, e.g., surface effects. While alternative non-destructive techniques like PFM indeed provide more direct and high-resolution domain wall imaging, a systematic comparison of different approaches would require a dedicated investigation. It is also not currently clear whether information obtained with a surface-sensitive technique can be directly applicable to the entire sample, given that in perovskites the surface effects are strong (including significant local inversion symmetry breaking; see e.g. ref. 40). We therefore consider this as an important direction for future research, but beyond the scope of the present manuscript.

To address the Referee's concerns regarding the potential effects of ion diffusion on the domain wall morphology, we performed additional time-dependent electrochemical staining experiments. Microscopic images taken before (0 minutes) and at 60, 90, 120, 180, and 240 minutes after the current began to flow through the sample were added to Section S2 of the revised Supplementary Information, along with a corresponding description. The results show that the structures formed at the onset of staining remain unchanged under repeated application of the electric field (Supplementary Fig. S3), indicating that ionic diffusion does not affect the intrinsic domain structure of MAPbBr_3 .

(4) The manuscript primarily focuses on the charge behavior at domain walls; however, since these domain walls originate from defects, which may involve defect recombination, the impact of domain wall dynamics on the overall stability of the material has not been addressed. It is recommended to investigate how the formation and annihilation states of domain walls influence the long-term stability and photoelectric conversion efficiency of the material.

We thank the Referee for raising this important question. Indeed, since domain walls constitute a class of lattice defects, their dynamics can have a direct effect on the structural composition of the sample as a whole, with the broadest possible implications. Nevertheless, although domain walls

can in principle be dynamically formed and annihilated, such processes appear to be strongly suppressed in practice. The chief reason is that domain walls belong to a special—topological—category of defects. As the Referee has rightly noted, a domain wall can only be annihilated via recombination with an “anti-wall” at the opposite side of the separating domain. This is an extremely intriguing possibility; however, we have not yet been able to observe such processes.

The topological nature of domain walls implies that their annihilation would entail macroscopic changes in the domain structure. It is difficult to imagine that such changes could be mediated purely by thermal fluctuations. Disregarding the obvious option of applying mechanical stress as falling beyond the scope of the present study of intrinsic properties of as-grown perovskites (complementary reports indicate strong flexoelectricity in MAPbBr₃ under stress, providing indirect confirmation of our general picture), one can instead consider influencing the domain structure through an external electric field. However, long-term biasing of MAPbBr₃ at moderate voltage (4 hours at 100 V mm⁻¹) did not reveal any noticeable change in the domain structure, as visualized by the dendrite patterns (see Supplementary Section S2 and Supplementary Fig. S3).

In a dedicated attempt to affect the domain structure through an applied voltage, we used the maximum field available to us in the laboratory (approximately 2000 V across a ~ 5 mm sample). In this experiment, we did observe a hint of macroscopic changes in the global birefringence structure of the sample that could be interpreted as field-induced modifications of the domain structure. This interpretation is further supported by the hysteretic nature of the phenomenon. However, we refrained from including these results in the present report, because the observations have not yet been conclusive.

To summarize, we agree that, in general, domain-wall dynamics is potentially an important factor that has to be thoroughly investigated. However, the existing evidence suggests that it becomes noticeable only under exceptional circumstances—such as extremely high electric fields—which are not normally encountered in typical perovskite practice.

(5) The current photocurrent measurements reflect the overall response between electrodes, which do not prove that the photo-generated carriers indeed undergo long-distance transport along specific domain wall pathways rather than through the bulk phase. More direct spatially resolved evidence is needed. It is recommended to attempt utilizing microscopic photocurrent imaging techniques (such as Laser Beam Induced Current Imaging - LBIC, or Scanning Photocurrent Microscopy - SPCM) to intuitively map the correlation between photocurrent generation hotspots and domain wall positions at the micrometer/sub-micrometer scale, and observe whether the current truly flows along the domain walls.

The Referee is right in noting that the photocurrent measurement reflects the overall response between the electrodes, which does not necessarily correspond to long-distance transport of charge carriers. However, this is also not our claim. As explained in detail in the Discussion section, the photocurrent we measure is a displacement current produced by rapid changes in the electric polarization of the sample after localized injection of charge carriers through nonlinear optical excitation. This includes (1) spatial separation of electrons and holes on opposite sides of the domain walls driven by local electric fields generated through the flexoelectric effect and (2)

subsequent recombination of charge carriers via tunneling through the flexoelectric potential barrier.

Since we do not apply bias to the sample and no carrier-type-selective electrodes are attached to the sample surfaces, no net charge transport along domain walls is involved. To confirm the displacement nature of zero-bias photocurrent, we have performed an additional experiment where we measured the net transfer of charge by zero-bias photocurrent (Supplementary Fig. S15). As described in Section S8 of the revised Supplementary Information, we observe that on average there is no net charge transfer, which is in full agreement with the physical picture described in the manuscript.

Our conclusion is that the domain walls exponentially suppress the recombination of photocarriers separated by the flexoelectric fields. Because the lifetime of these charges can reach millisecond time scales, they may diffuse across micrometer-scale distances before recombination. In dedicated devices equipped with charge-selective electrodes, such diffusion-driven transport would manifest as a photovoltaic effect. The central issue addressed by our paper is therefore the mechanism by which—despite reports of extremely short exciton recombination times—such long carrier lifetimes can arise, enabling long-range diffusion transport as an indirect consequence.

(6) In real polycrystalline perovskite thin films, in addition to domain walls, grain boundaries, surfaces, defect states, etc., may all serve as pathways or traps for carrier transport. The manuscript needs to delve more deeply into the competitive relationship between the discovered "flexoelectric domain wall channels" and the intrinsic transport pathways, which is crucial for evaluating their practical device application value.

We thank the Referee for this insightful comment. We fully agree that in real polycrystalline perovskite thin films, grain boundaries, surfaces, and defect states coexist with domain walls and can significantly impact charge transport. A systematic comparison of the relative contributions and competition between various intrinsic transport pathways is therefore essential for evaluating the performance of practical devices, such as solar cells. However, in this work, our focus is intentionally on a bulk single-crystal MAPbBr₃, where the effects of grain boundaries and surfaces are minimized. This approach allowed us to focus on the basic aspects of the novel phenomenon and isolate and identify the role of flexoelectric domain walls in the spatial separation of charge carriers, extending their lifetime and diffusion lengths. We therefore consider expanding the study to polycrystalline thin films as an important direction for future research, but beyond the scope of the present manuscript.

(7) The manuscript proposes the potential application of flexoelectric domain walls in perovskite solar cells, yet it does not thoroughly explore the commercialization pathways and associated challenges. It is recommended to add a dedicated section discussing how this discovery can drive further advancement in perovskite solar cell technology, including an analysis of potential market barriers, scalability issues, cost-effectiveness considerations, and strategies for overcoming technical hurdles in the transition from laboratory research to commercial products.

We agree with the Referee that analyzing commercialization pathways and associated challenges—such as scalability, cost-effectiveness, and market barriers—is important for

evaluating both the potential impact of our findings on perovskite solar cell technology and the economic feasibility of translating reported fundamental discoveries into practical applications. However, the main goal of the present work is to understand the basic physical mechanisms behind complex charge-carrier dynamics in lead-halide perovskites and to demonstrate the role of flexoelectric domain walls in a model system (MAPbBr_3). A detailed assessment of commercialization strategies is therefore postponed for future work.

Reviewer #1 (Remarks to the Author):

While the authors have addressed several reviewer questions, some points remain unclear in the current manuscript. First, why did the authors choose MAPbBr₃ instead of CsPbBr₃, which adopts a perfectly cubic structure at room temperature?

We thank the Referee for this question. We would like to clarify that CsPbBr₃ does not adopt a cubic structure at room temperature. Structural studies (e.g., doi: 10.1143/JPSJ.37.1393, doi: 10.1023/A:1022836800820) consistently show that CsPbBr₃ is orthorhombic at room temperature, with phase transitions to the tetragonal and cubic phases occurring at approximately 88°C and 130°C, respectively.

Could the authors provide similar experimental results on CsPbBr₃ crystals? This comparison would help clarify whether the proposed method for understanding the flexoelectric effect in perovskites is general or specific to the organic–inorganic hybrid system.

We appreciate the suggestion. However, CsPbBr₃ is cubic only above ~130°C, whereas MAPbBr₃ begins to decompose already near 80°C, making direct comparison between the two systems experimentally impossible under comparable conditions. Addressing the question of the role of A-site cation by comparing room-temperature single-crystal MAPbBr₃ to single-crystal CsPbBr₃ in its high-temperature cubic phase would therefore primarily involve exploring the role of thermal fluctuations of lead-halide lattice on phenomena like ferroelasticity, domain structure, and charge transfer. This would constitute a substantial and separate study beyond the scope of the present work.

Furthermore, since MAPbBr₃ is not a representative material for state-of-the-art photovoltaic devices, the authors should explain how the present study offers new insights or fundamental understanding relevant to more practical photovoltaic compositions, such as mixed-cation lead iodide perovskites.

We thank the Referee for raising this important point. The objective of the present study is to elucidate the microscopic origins of exceptionally long photocarrier recombination times in lead-halide perovskites (LHPs) in general. The complexity of mixed-cation/mixed-halide compositions can obscure the underlying physics, therefore we find it essential to begin with a well-characterized prototypical system.

MAPbX₃ (X = Cl, Br, I) perovskites are widely used as model systems for fundamental optical, electronic, and defect-physics studies in LHPs. Previously, several works on MAPbI₃ [14,15,23,24] have suggested possible non-centrosymmetry in its room-temperature tetragonal phase. For this reason, we intentionally selected a composition - MAPbBr₃ - whose lattice structure is unambiguously cubic and centrosymmetric at room temperature, ensuring that any observed bulk photovoltaic response could not be attributed to intrinsic symmetry breaking. Additionally, this compound offers favorable optical properties and the ability to visualize domain walls electrochemically - an essential aspect of our study. Finally, the fact that MAPbBr₃ is cubic at room temperature enables study of pristine samples that are naturally grown into cubic phase and

in between the synthesis and measurement stages have not undergone any structural phase transitions that can potentially compromise sample integrity.

With regard to the broader relevance of our results for practical photovoltaic systems and for LHPs in general, we summarize below - for convenience - where this is discussed in the revised manuscript:

- *Lines 266–280, 316–318, 337–340:* Flexoelectric domain walls naturally account for the long photocarrier recombination times characteristic of LHPs, a key factor in their photovoltaic performance [42].
- *Lines 225–231:* The framework provides a coherent explanation for hysteretic polarization [14,15] and pyroelectric effects [16] observed in nominally centrosymmetric LHPs.
- *Lines 232–240:* We apply the framework to MAPbI₃ and show that its long-debated ferroelectric-like behavior can be reconciled with localized flexoelectricity, consistent with both hysteretic polarization [14,15] and the absence of SHG in bulk [37,39,40].
- *Lines 240–243:* We describe how domain-wall flexoelectricity offers a general microscopic mechanism for local inversion symmetry breaking across LHPs.
- *Lines 316–321, 337–343:* We highlight the importance of mesoscale conductive pathways for long-range charge transport.
- *Lines 322–329:* We discuss potential implications of flexoelectric domain walls for photovoltaic design, including both the benefits for charge separation and the trade-offs under high illumination.

Finally, we revised the Discussion section to explicitly note that mesoscale structural engineering based on domain-wall phenomena may offer a complementary strategy to compositional optimization for improving photovoltaic performance.

Regarding Ag doping in MAPbBr₃, what role does Br⁻ ion migration play in defect formation induced by Ag incorporation? Could such halide ion migration contribute to the observed flexoelectric effect and local symmetry breaking?

We thank the Referee for these questions. As noted in our response to Question 8 in the previous round, all optical, photocurrent, and X-ray measurements relevant to domain-wall flexoelectricity were performed on pristine, undoped MAPbBr₃ crystals. We further clarified this information in the revised manuscript to avoid possible confusion.

Ag-doping was used exclusively for electrochemical staining to visualize domain structures inferred from polarized microscopy. A detailed investigation of Br⁻ migration and Ag-induced defects under an applied field is therefore outside the scope of this work. Assessing possible ion-migration-related contributions to flexoelectricity in doped samples would require a dedicated follow-up study focused specifically on Ag-doped MAPbBr₃.

Lastly, in response to Reviewer 4, the authors mention that the preservation of inversion symmetry in bulk perovskites has been confirmed by optical SHG experiments (Refs. 37 and 38). However, upon reviewing these references, the materials studied are not MAPbBr₃.

We appreciate the opportunity to clarify this point. Both cited works report SHG measurements on MAPbBr₃. In Ref. 37 (p. 5), the authors state: “...as a reference sample we took the measurement on single crystals of MAPbBr₃ with cubic crystal structure and did not detect any SHG signal...” Similarly, Ref. 38 reports (Fig. 2 and p. 3): “The spectra show only odd harmonics...; the absence of even harmonics originates from the inversion symmetry of the perovskite crystal structure.” Because the experiments in Ref. 38 used excitation at 0.35 eV, the second harmonic (0.70 eV) lies well within the transparency range of MAPbBr₃ (e.g., doi: 10.1021/acs.jpcc.5b07432, doi: 10.1021/acs.jpcclett.3c01158), so the stated absence of *all* even harmonics necessarily includes the second harmonic.

I do not believe that MAPbBr₃ exhibits any SHG response, as its centrosymmetric structure should preclude second-order nonlinear optical activity. To my knowledge, no SHG signal has been reported for this material.

We completely agree with the Referee. Indeed, the absence of SHG in MAPbBr₃ reflects its *intrinsic* centrosymmetry. This observation forms the basis of our conclusion that the zero-bias photocurrent arises from *extrinsic* phenomena associated with localized flexoelectricity at domain walls.